# Effects of Soaking on the Volatile Compounds, Textural Property, Phytochemical Contents, and Antioxidant Capacity of Brown Rice

**DOI:** 10.3390/foods11223699

**Published:** 2022-11-18

**Authors:** Ling Zhu, Chengtao Yu, Xianting Yin, Gangcheng Wu, Hui Zhang

**Affiliations:** 1School of Food Science and Technology, Jiangnan University, Wuxi 214122, China; 2National Engineering Research Center for Functional Food, Collaborative Innovation Center of Food Safety and Quality Control in Jiangsu Province, Jiangnan University, Wuxi 214122, China

**Keywords:** brown rice, soaking, physiochemical properties, nutritional property

## Abstract

Brown rice is a staple whole grain worldwide. Hence, the effects of cooking on the nutritional properties of brown rice are important considerations in the field of public health. Soaking is a key stage during rice cooking; however, different rice cookers use different soaking conditions and the effects of this on the physiochemical properties and nutritional composition of cooked brown rice remain unknown. In this study, the setting of varied soaking conditions was realized by a power-adjustable rice cooker, and the effects of soaking temperature (40, 50, 60 and 70 °C) and time (30 and 60 min) on cooked brown rice were thoroughly analyzed. Textural results revealed that cooked brown rice was softer and stickier after soaking. Grain hardness decreased by increasing the soaking temperature and time. Furthermore, stickiness after soaking for 60 min was higher than that after 30 min, and this decreased with the soaking temperature. There was no significant unpleasant flavor after soaking, and the volatile compound profile between soaked and unsoaked brown rice was similar. Neither soaking temperature nor time had any significant effect on the phytochemical contents (phenolic compounds, α-tocopherol and γ-oryzanol) or antioxidant capacity of cooked brown rice, whereas γ-aminobutyric acid content was effectively preserved within a certain soaking temperature range. Textural properties can be effectively controlled by soaking temperature and time, and nutritional properties remain stable when soaking at 40–70 °C for 30–60 min.

## 1. Introduction

Whole cereal grains have health benefits, especially because of the lower risk of chronic diseases associated with their consumption [1]. For many years, white or polished rice has been the predominant form of rice consumed. However, with increasing public awareness about the nutritional imbalance (more than 90% starch in dry solids) caused by the consumption of white rice, the quest for healthier alternative forms of rice has become a research hotspot in food science. Brown rice retains partial bran layers and the embryo, which is more nutritious but has a bitter taste and takes longer to cook. Greater amounts of phytochemicals exist in the layers, such as phenolic compounds, α-tocopherol, and γ-oryzanol, which exhibit antioxidant, neuro-protective, antidiabetic, and antihypertensive effects [2]. In general, germination treatment can soften cooked brown rice grains and induce the formation of γ-aminobutyric acid (GABA), which is useful for improving the nutritional quality [2]. However, germinated brown rice requires sterilization and preservation, which is time-consuming and expensive. Therefore, it is necessary to find an efficient way to improve the texture and maintain the nutritional value of brown rice.

For years, different heating methods or devices such as open fire, electrical, steam cooking, induction ovens and microwaves have been used for rice cooking [3]. Among these methods, open pan cooking is still used in developing countries; pressure and steam heating are often used for shorting the heating time and softening the texture of paddy and brown rice [4]. However, due to safety and conventional issues, their pressure size and heating medium are limited. A better option for large-scale cooking would be electrical heating due to its convenience and continuous cooking process [5]. In most households, the use of an electric rice cooker for cooking rice has become very popular. Particularly, the induction heating cooker is a kitchen appliance that has been widely adopted owing to its fast-heating capacity and accurate heating control. The cooking procedure or curve is the main characteristic of any rice cooker, as it determines the appearance and texture of cooked rice. Compared to direct heating, a multi-stage cooking protocol, especially one that involves soaking before cooking, might improve the cooking quality of both white rice and brown rice [3]. Compared to that of white rice, the soaking temperature of brown rice is usually higher, whereas the cooking time is longer [6,7]. Furthermore, the rice soaking conditions used with cookers of different brands vary considerably and contribute the most to the differences among the corresponding cooking curves. However, reports on how soaking or cooking conditions influence the final textural properties of cooked rice are scarce at best. Furthermore, the relationship between soaking conditions and rice flavor and whether the selected soaking temperature or time has an effect on bioactive components are still unclear.

Texture and volatile flavor are two critical factors affecting consumer acceptance [8]. Instead of sensory evaluation by trained panelists, instrumental measurement of texture is often used to evaluate the mechanical characteristics of foods during the oral processing [9]. The texture profile analysis (TPA) has been used to develop the understanding texture properties of different foods and the texture terms are classified into five basic parameters. For cooked rice, hardness and stickiness are two key indexes for palatability evaluation [10]. Phytochemicals present in the grain and antioxidant capacity are the two major advantages of brown rice from a health standpoint [11]. Furthermore, presoaking, which involves soaking at a low temperature for hours and then removing the soaking water before cooking (refill water), could improve the viscoelasticity and promote flavor release in cooked rice [12]. Unfortunately, these soaking conditions (long time soaking and refilling water) are not suitable when using a rice cooker for convenience and effective cooking [5]. Indeed, an extended soaking period might cause microbial contamination and an unpleasant flavor, especially in brown rice [13,14]. Therefore, a thorough comparison is necessary to determine how soaking conditions during cooking affect the properties of brown rice. More sensitive cooking devices and more quality comparison information under uniform hardware conditions are also needed to optimize the cooking curves to obtain better quality of cooked brown rice.

Here, we report, to our knowledge, the first study using a power-adjustable rice cooker, for which different soaking conditions can be set to compare different cooking curves for the same cooker. The texture, volatile compounds, phytochemicals, and antioxidant capacity of brown rice cooked under different soaking conditions were analyzed and compared. Our study tries to provide a theoretical foundation to optimize automated brown rice-cooking protocols.

## 2. Materials and Methods

### 2.1. Materials

A *japonica* brown rice strain, with a milling degree of approximately 3%, was cultivated in Jiling Province, China, for use in the experiments reported herein. Chemicals were prepared as follows: TCI Corporation (Tokyo, Japan): 2,4,6-trimethylpyridine (internal standard); Sigma-Aldrich (Burlington, MA, USA): C7–C30 alkane standards, α-tocopherol; Yuanye Biotechnology Co., Ltd. (Shanghai, China): gallic acid, ABTS (2, 2′-azino-bis(3-ethylbenzothiazoline-6-sulfonate acid)); Beijing Bailingwei Technology Co., Ltd. (Beijing, China): TPTZ (2,4,6-tris(2-pyridyl)-s-triazine), and Trolox (6-hydroxy-2,5,7,8-tetramethylchromane-2-carboxylic acid), individual phenolic acid standards; Toronto Research Chemicals (North York, ON, Canada): γ-oryzanol standards.

### 2.2. Soaking Conditions during Cooking

Based on the parameters of different brands of rice cookers, the soaking temperature was set to 40, 50, 60, or 70 °C, and soaking time was 30 or 60 min. All soaking and cooking combination treatments (heating rate, temperature, and time, etc.) were set using an adjustable electric rice cooker, which was assembled by an electric appliance company (Joyoung, Hangzhou, China). Briefly, 400 g of brown rice and 800 mL distilled water were placed into the adjustable rice cooker. Then, eight cooking processes with different soaking temperature (40, 50, 60 and 70 °C) and time (30 and 60 min) were used. After soaking, the temperature increased to 100 °C and maintained for 40 min. Unsoaked samples were directly heated from 25 °C to a boiling temperature and maintained for 40 min as the control. The heating rate of soaked rice samples was 9 °C/min, and that of unsoaked rice was 4 °C/min. The chosen heating rate for both soaked and unsoaked rice samples were chosen according to the usual condition applied by rice cookers on the market.

Some cooked brown rice samples were freeze-dried and milled into flour (180 µm mesh) for further analysis. In the following sections, samples are designated based on the soaking temperature and time combination treatments. For example, “40-30” indicates rice grains soaked at 40 °C for 30 min.

### 2.3. Texture Analysis

Cooked brown rice (approximately 5 g) was cooled at 25 °C for 20 min. The cooled rice sample was filled in aluminum cups (internal diameter was 3.5 mm), and then the texture was measured with a texture analyzer (TA-XT plus, stable Micro Systems Ltd., Godalming, UK). The two-cycle compression program (repeated 12 times) was conducted using a P/25 cylindrical probe. The cooked rice samples were compressed to 50% deformation at a pre-load test and post-test speed of 1.0 mm/s [5].

### 2.4. Analysis of Volatile Compounds

Volatile compounds were analyzed as follows: Briefly, 15 g of cooked brown rice was placed in a 50 mL headspace vial and equilibrated at 60 °C for 10 min. Then, A DVB/CAR/PDMS fibre (1 cm, 50/30 µm, divinylbenzene/carboxen/polydimethylsiloxane) was used to extract volatile compounds at 60 °C for 30 min [15]. The desorption temperature and time were 250 °C and 5 min, respectively. The separation of volatile compounds was using a DB-Wax capillary column (30 m × 0.25 mm × 0.25 µm). The flow rate of carrier gas (highly purified helium) was 1.0 mL/min. The temperature program was initially maintained at 40 °C for 3 min, then increased to 150 °C by 3 °C/min, further increased to 240 °C by 10 °C/min and held at 240 °C for 4 min. The ionization mode was electron impact at 70 eV, and the scan range was 35–450 m/z. Next, 2,4,6-trimethylpyridine was added as an internal standard for semi-quantification. The qualitative analysis of different volatile flavor compounds (such as hydrocarbons, aldehydes, furans, alcohos, ketone and others) was according to the NIST database (National institute of standards and technology), matching and comparison of normal alkanes for retention coefficient calculation [16].

### 2.5. Moisture Content

The collected cooked grain (3.0 g) and flour samples (1.0 g) were dried to a constant weight (105 °C for 2–6 h) [17]. The moisture results were calculated based on ratio of moisture difference before and after drying to fresh weight.

### 2.6. Analysis of Phenolic Compounds

Phenolic compounds in cooked brown rice were extracted and measured as follows. Firstly, a mixture of freeze-dried sample (1.0 g) and 80% ethanol (20 mL) was shaken at 25 °C for 2 h, followed by alkali hydrolysis (4 mol/L, 20 mL) under an N_2_ atmosphere for 1 min, shaking for 4 h [18]. UPLC-PDA-MS/MS was used to identify and quantify the phenolic compounds (Waters Co., Milford, MA, USA). Two mobile phases with a flow rate of 0.4 mL/min (A, 0.1% formic acid in acetonitrile; B, 0.1% aqueous formic acid) were used for gradient elution (B, 20 min: 95% to 50%; 20.10 min: 50% to 95%; 22 min: 100% to 0%; 22.10 min: 0% to 95%; 30 min 95% to 95%) and sample separation. The column type was an ACQUITY UPLC HSS T3, the size was 2.1 × 100 mm, 1.8 µm, and the temperature was 35 °C. The detection wavelength was set at 280 and 320 nm. For mass spectrometry conditions, the ionization mode was positive, and the ion source temperature was maintained at 120 °C. The cone energy and collision energy were 20–40 eV and 10–22 eV, respectively [19]. The content of detected phenolic compounds was recorded in units of nanograms per gram of dry weight (ng/g DW).

### 2.7. Analysis of Nutritional Components

α-Tocopherol and γ-oryzanol contents: Two grams (dry weight) of cooked brown rice flour was mixed with 20 mL of methanol and vortexed for 2 min. Then, samples were centrifuged at 6000× *g* for 5 min, and the supernatant was separated and collected for use. Then, 50 µL extracts were injected into a HPLC instrument (Shimadzu, Kyoto, Japan). The column was an SPD-20A photodiode array and an ACQUITY UPLC HSS T3 column (4.6 × 250 mm, 5 µm) [20]. The mobile phase of α-tocopherol was water:methanol = 5:95 (*v*:*v*), whereas that of γ-oryzanol was acetonitrile:methanol = 20:80 (*v*:*v*), with a rate at 1.0 mL/min. The wavelengths of detection were 295 nm and 325 nm, respectively. The HPLC column system was maintained at 35 °C.

GABA content: The extraction of GABA was first conducted by mixing 1 g (dry weight) with 25 mL trichloroacetic acid (5 g/100 mL). Following this, all solution samples were shaken at 250 rpm for 2 h and let stand for another 12 h (25 °C) [15]. The centrifugal supernatant (15,000 r/min for 30 min) of the extracted GABA was collected. Detailed information about HPLC analysis parameters and methods are provided in the Appendix A.

### 2.8. Analysis of Antioxidant Capacity

ABTS working fluid was produced by diluting the ABTS^•+^ (mixed 200.0 mg ABTS with 34.4 mg potassium persulfate in 50 mL distilled water; it was allowed to stand in the dark for 12 h) radical cation with methanol to 0.700 ± 0.020 at 734 nm. Then, taking 5 mL of ABTS working fluid mixed with 0.25 mL of extracted samples, the mixture was allowed to stand for 30 min, and the absorbance was measured as indicated previously [21].

Ferric ion reducing capacity was measured as follows: 10 mmol/L Fe^3+^-tripyridine triazine (TPTZ, 40 mmol/L hydrochloric acid as solvent), 20 mmol/L FeCl_3_·6H_2_O solution, and 0.3 mol/L sodium hypochlorite (1:1:10, *v*:*v*:*v*) were mixed into iron reducing reagent (3.9 mL) and mixed with 0.1 mL of extracted sample, and the absorbance was measured at 593 nm [19,21].

## 3. Results and Discussion

### 3.1. Effects of Soaking Conditions on Textural Properties of Cooked Brown Rice

Hardness and stickiness are two important indices used to describe the texture of cooked brown rice [22]. Hardness could reflect the compactness of the internal structure of rice, which was gradually decreased (unsoaked rice, 2053.2 ± 87.4 g) with increasing soaking temperature and time (Figure 1). A higher soaking temperature or a longer period of soaking could promote greater water absorption by the endosperm throughout the outer bran layer [23]. The increased moisture content and its even distribution could facilitate the granules swelling and gelatinization of starch upon heating, and thus exhibit a lower hardness of cooked rice [23,24].

After soaking, the stickiness of brown rice grains increased from 177.6 ± 23.2 g·s to 255.6 ± 25.9~397.9 ± 20.6 g·s. Furthermore, the comparison of different soaking temperatures and time showed that the stickiness correlated positively with the soaking time (at the same soaking temperature) and negatively with the soaking temperature (at the same soaking time) (Figure 1). This result differs slightly from data reported for cooked white rice [7]. During heating or soaking, solids containing starch, protein, and some small-molecule substances leach from the surface and cracks of the rice grains into the water used for soaking and are then adsorbed onto the cooked grains at higher temperatures (80~100 °C), thereby creating stickiness [7]. As it is generally accepted, the stickiness of cooked rice relates with the amylopectin content of the leached solid [25]. The higher amount of amylopection, especially short amylopectin chains in the leachate, the greater opportunity for molecular chain interaction or bonding. Thus, more force is needed to make the cooked rice come apart, exhibiting higher stickiness [23]. In contrast, in brown rice, the retained cortex and aleurone layer of the grain maintain the endosperm in a tightly sealed state, thus largely reducing the leaching of endosperm starch into the soaking water. A lower soaking temperature corresponds with a lower water migration rate. More water was distributed on the surface than in the inner area of brown rice grains, causing a greater water gradient between the surface and inner region. For this reason, some wider internal cracks formed when they were soaked at low temperatures (30–50 °C, 5–30 min), compared with high soaking temperatures [26]. Thus, at higher soaking temperatures (60–70 °C), the stickiness decreased as compared with low temperatures. Overall, soaking during cooking could provide a softer and stickier texture of cooked brown rice and vary with different soaking conditions.

### 3.2. Comparison of Volatile Profiles of Cooked Brown Rice

As mentioned in the literature, germinated brown rice might develop undesirable flavors when soaked at room temperature for hours [13]. This change limits the edibility of brown rice, mainly owing to the activation and release of enzymes, such as fat oxidation by lipases and hydrolysis of starch and protein by amylases, and proteases, respectively, during germination. However, it is still not clear whether short-term soaking in a rice cooker would also produce different volatile compounds or undesirable flavors. Thus, the main profiles of volatile compounds with different soaking temperatures and times were captured and compared (Table 1). As can be seen, 28 volatile compounds in unsoaked brown rice were also detected in all soaked brown rice samples. The contents of these volatile compounds were similar and no new flavor substances were produced, indicating that a certain degree of soaking did not introduce undesirable flavors.

Furans accounted for 41.31~43.29% of the total volatile compounds, followed by aldehydes, which accounted for 35.4~38.33% (Figure 2). The odor threshold of aldehydes and some furans was low, thereby contributing to the main aroma of brown rice (floral, plant-like, and sweet) [27]. In cooked samples, the most abundant volatiles were hexanal and 2-pentylfuran. Hexanal was the major contributor of an off-flavor (similar to oxidized oil) at high concentrations [27,28]. Similarly, 2-pentylfuran also produced an unpleasant green, beany, and earthy aroma at high concentrations [29]. Additionally, 2, 3-dihydrobenzofuran gave an undesirable soybean odor [30]. By comparison, there was no significant difference in volatile compounds among soaking conditions (Table 1 and Figure 2). The results indicated that these soaking conditions set in the cooking procedure of the rice cooker had no adverse effects on the sensory quality of cooked brown rice. According to a previous study, volatile compounds are mainly volatilized during the heating and boiling stages [31]. After soaking, the release of volatile compounds combined with the high temperature induced Maillard reaction and thermal decomposition provided cooked rice with an intense volatile flavor [16]. Therefore, the volatilization of volatile compounds in brown rice was unaffected, although they and/or their precursors seemingly leached into the water during soaking.

### 3.3. Comparison of Phenolic Profile of Cooked Brown Rice

Polyphenols in brown rice are composed of free and bound phenols with different bioavailabilities and beneficial health effects. The dietary intake of bound phenol might have chemo-preventive activity against colon cancer, whereas free phenol is easily absorbed into bodies during digestion, consequently inhibiting the oxidation of low-density lipoproteins, cholesterol, and liposomes [32]. The activity of phenolic compounds was closely related to the temperature and water content. The effect of soaking on phenolic profiles of cooked brown rice is shown in Table 2. There were four phenolic compounds present in the samples analyzed, namely ferulic acid, isoferulic acid, *p*-coumaric acid, and salicylic acid. Among the four, the content of ferulic acid, mainly in the bound form, was the highest, accounting for 61.33~66.03% of total polyphenols, followed by isoferulic acid (16.20~19.43%) and *p*-coumaric acid (14.93~15.97%).

In raw rice, the total contents of free and bound phenol were 6.43 and 81.73 ng/g DW, respectively. After cooking, these values decreased to 2.60~2.80 and 63.13~66.05 ng/g DW, respectively. Thus, the extent of the decrease was 56.5~59.6% and 19.2~22.8%, in each case, in agreement with previous studies [33,34]. It has been reported that a high temperature during cooking causes the oxidative degradation of polyphenols [9,35]. Furthermore, owing to the leaching of phenolic compounds, soaking before cooking for hours can also reduce the polyphenol content in cereals [36]. Thus, it was proposed that cooking without pre-soaking might maximize the polyphenol contents. However, in this study, we found no significant differences among different soaking conditions. The results indicated that soaking during cooking had no influence on the phenolic composition or contents in cooked brown rice; the main reason being that, in contrast with removing water after soaking (germinated brown rice), water may be reabsorbed into brown rice grains when cooked in a rice cooker [37]. Thus, high-temperature soaking might even facilitate phenolic compound leaching, whereas the reabsorption of water during subsequent cooking could prevent the loss of phenolic compounds.

### 3.4. Effect of Soaking on Main Nutritional Components in Cooked Brown Rice

As shown in Figure 3A, the initial GABA content in brown rice was 29.76 ± 1.28 mg/100 g DW, but this decreased to 16.44 ± 0.34~23.34 ± 0.74 mg/100 g DW after cooking. The results showed that there was a significant loss of GABA during cooking, in agreement with the results of a previous study [38]. GABA content was similar across soaking time treatments; however, it differed with soaking temperature. The retention of GABA was relatively high when brown rice was soaked at 60 °C, compared to that with the other soaking temperature treatments. This result might be explained by the protein hydrolysis-induced increase in free amino acids, such as GABA and serine, within a certain temperature range [39].

α-Tocopherol and γ-oryzanol are the two unique components contained in brown rice (mainly in the bran layer) [40]; compared with those in uncooked brown rice, their contents in soaked brown rice decreased by 57.2~61.9% (α-tocopherol) and 11.1~15.1% (γ-oryzanol) (Figure 3B). The results indicated that γ-oryzanol in brown rice remained relatively stable upon cooking, compared with α-tocopherol. However, there were no significant differences in α-tocopherol or γ-oryzanol contents among cooked samples soaked under different conditions, suggesting that α-tocopherol and γ-oryzanol contents were mainly affected by heating and boiling stages during cooking rather than soaking before cooking.

### 3.5. Comparison of Antioxidant Capacity of Cooked Brown Rice

Antioxidant properties of phenolic compounds determine their health benefits, such as their well-known anti-diabetes effects. In general, different measuring principles and reaction conditions make the results of different methods have some differences. Therefore, ABTS and FRAP, the two antioxidant capacity evaluation methods, were selected for the comparisons. The changes in the antioxidant capacity of rice samples measured by ABTS under different soaking conditions were in line with that measured by FRAP, which showed that both cooking and soaking had an effect on the antioxidant capacity of brown rice (Figure 4). For intact brown rice, the total antioxidant capacity based on ABTS and FRAP assays was 259.33 ± 9.45 and 240.14 ± 8.04 mg TE/100 g DW, respectively. In contrast, the corresponding values for cooked brown rice samples ranged from 207.37 ± 5.04 to 211.16 ± 4.87 and from 174.98 ± 4.39 to 180.18 ± 6.13 TE/100 g DW, respectively. Thus, total antioxidant capacity of cooked samples decreased by 18.6~20.0% and 25.0~27.1%, for ABTS and FRAP assays, respectively. The results indicated that rice cooking (without soaking) decreased the antioxidant capacity of brown rice. The decrease in antioxidant capacity was mainly reflected by an overall loss of phenolic compounds [41]. Furthermore, most of this reduction was associated with the loss of the free phenolic fraction (Figure 4).

In comparison, the free, bound, and total antioxidant capacities of soaked brown rice were significantly lower than these of intact brown rice, but the difference between unsoaked and soaked brown rice was not distinct. Meanwhile, the differences in antioxidant capacity among samples treated under different soaking conditions were not statistically significant. The results indicated that the effects of soaking on antioxidant capacity were limited, compared to those associated with the cooking process. Thus, by adjusting the soaking conditions, the cooking process can be optimized, thereby preserving the nutritional value of cooked brown rice, while concomitantly providing a better texture.

## 4. Conclusions

A study on the effects of soaking conditions on the texture, contents of volatile compounds, phenols, and antioxidation of cooked brown rice was performed by a powder-adjustable rice cooker. Soaking provided a softer and stickier texture of cooked brown rice. The hardness of cooked brown rice decreased by increasing the soaking temperature and time, and stickiness after soaking for 60 min was higher than that after 30 min. Moreover, soaking at 60 °C best preserved GABA closer to the initial content. Additionally, soaking during cooking had no marked effect on the volatile or phenolic compound profiles, α-tocopherol content, γ-oryzanol content, or antioxidant capacity of cooked brown rice. These results expand our knowledge of the effects of soaking conditions during cooking on the sensory and nutritional quality of brown rice. This study helps us and manufacturers to drive the cooking process using a set temperature/time in a rice cooker to obtain an optimal textural, phytochemical and antioxidant capacity in brown rice.

## Figures and Tables

**Figure 1 foods-11-03699-f001:**
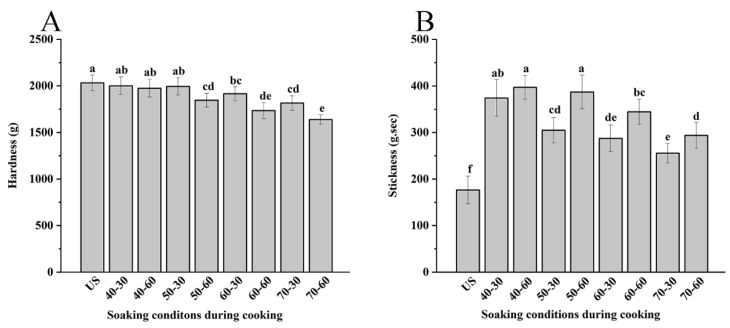
The hardness (**A**) and stickiness (**B**) of cooked brown rice under different soaking conditions. “US” respected unsoaked brown rice; soaking conditions such as “40-30” respected soaking at 40 °C for 30 min. Results with different lowercase letters are significantly different (*p* < 0.05).

**Figure 2 foods-11-03699-f002:**
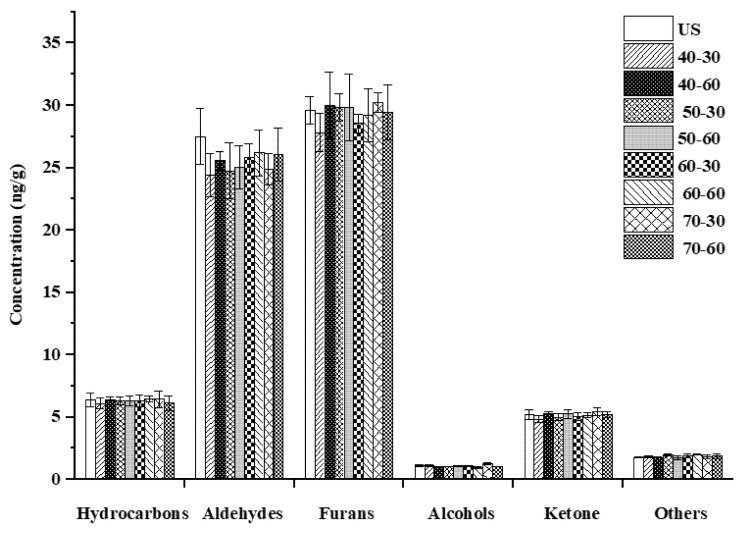
The concentrations of different chemical groups of volatile compounds of cooked brown rice under different soaking conditions. “US” respected refers to brown rice; soaking conditions such as “40-30” refer to soaking at 40 °C for 30 min.

**Figure 3 foods-11-03699-f003:**
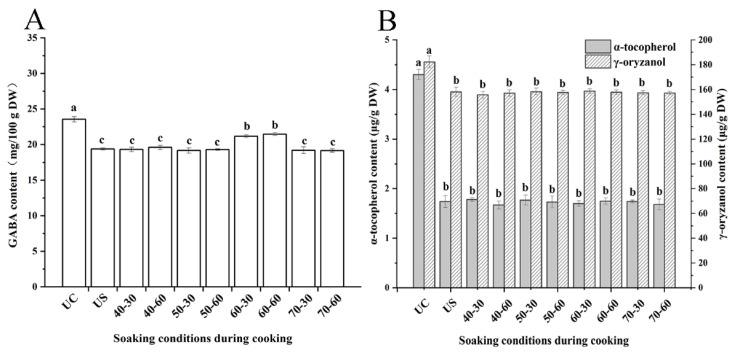
The GABA (**A**), γ-oryzanol and α-tocopherol (**B**) contents of uncooked brown rice and cooked brown rice under different soaking conditions. Abbreviation was listed: UC (uncooked brown rice), US (unsoaked brown rice); soaking conditions such as “40-30” respected soaking at 40 °C for 30 min. Results with different lowercase letters are significantly different (*p* < 0.05).

**Figure 4 foods-11-03699-f004:**
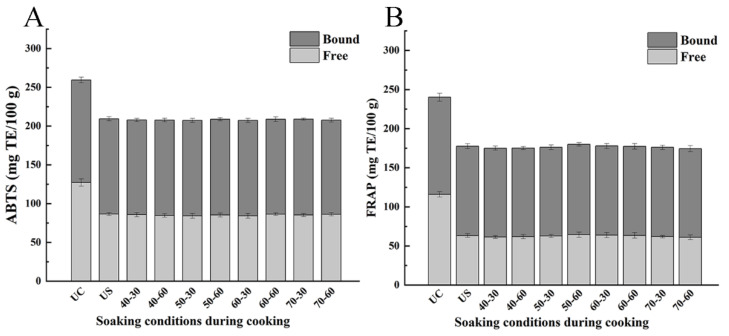
The antioxidant capacities determined by ABTS (**A**) and FRAP (**B**) assays of uncooked brown rice and cooked brown rice under different soaking conditions. Abbreviation was listed: UC (uncooked brown rice), US (unsoaked brown rice); soaking conditions such as “40-30” respected soaking at 40 °C for 30 min.

**Table 1 foods-11-03699-t001:** Volatiles profile of cooked brown rice under different soaking conditions during cooking.

Compound (ng/g)	Unsoaked	40-30	40-60	50-30	50-60	60-30	60-60	70-30	70-60	CRI	LRI
Hydrocarbons											
Octane	2.27 ± 0.27	2.18 ± 0.11	2.45 ± 0.22	2.31 ± 0.18	2.39 ± 0.30	2.38 ± 0.23	2.29 ± 0.28	2.31 ± 0.10	2.16 ± 0.25	798	800
α-Pinene	0.61 ± 0.09	0.53 ± 0.13	0.43 ± 0.05	0.46 ± 0.02	0.55 ± 0.09	0.49 ± 0.07	0.58 ± 0.11	0.65 ± 0.05	0.52 ± 0.07	1032	1029
4-Methylene-1-(1-methylethyl)- bicyclo[3.1.0]hexane	3.45 ± 0.19	3.26 ± 0.25	3.31 ± 0.33	3.35 ± 0.15	3.21 ± 0.07	3.29 ± 0.20	3.47 ± 0.23	3.38 ± 0.36	3.27 ± 0.28	1109	1116
Pentylcyclopropane	0.15 ± 0.03	0.16 ± 0.01	0.15 ± 0.02	0.14 ± 0.03	0.17 ± 0.02	0.17 ± 0.03	0.13 ± 0.01	0.14 ± 0.02	0.18 ± 0.01	1538	NF
Aldehydes											
Hexanal	12.24 ± 1.39	10.88 ± 0.98	11.34 ± 0.33	11.60 ± 1.24	11.94 ± 0.89	13.18 ± 0.85	12.02 ± 0.92	11.27 ± 0.98	12.44 ± 1.34	1077	1080
Octanal	3.70 ± 0.24	3.64 ± 0.14	3.74 ± 0.34	3.54 ± 0.35	3.42 ± 0.38	3.48 ± 0.30	3.31 ± 0.07	3.58 ± 0.10	3.25 ± 0.22	1280	1284
Nonanal	9.44 ± 0.52 ^a^	7.51 ± 0.86 ^b^	8.32 ± 0.55 ^ab^	7.24 ± 0.76 ^b^	7.69 ± 0.46 ^b^	6.96 ± 0.28 ^b^	8.25 ± 0.65 ^ab^	7.77 ± 0.20 ^b^	8.38 ± 0.42 ^ab^	1372	1390
(E)-2-Octenal	0.37 ± 0.06 ^b^	0.45 ± 0.07 ^ab^	0.32 ± 0.03 ^b^	0.40 ± 0.06 ^b^	0.41 ± 0.05 ^b^	0.49 ± 0.08 ^ab^	0.58 ± 0.09 ^a^	0.48 ± 0.03 ^ab^	0.38 ± 0.02 ^b^	1410	1416
Decanal	0.27 ± 0.06	0.32 ± 0.07	0.23 ± 0.03	0.28 ± 0.02	0.24 ± 0.04	0.22 ± 0.03	0.33 ± 0.06	0.29 ± 0.08	0.24 ± 0.06	1477	1484
Benzaldehyde	1.15 ± 0.19	1.21 ± 0.20	1.27 ± 0.17	1.39 ± 0.07	1.07 ± 0.07	1.21 ± 0.06	1.38 ± 0.04	1.23 ± 0.11	1.12 ± 0.09	1500	1508
(E)-2-Nonenal	0.31 ± 0.04 ^ab^	0.40 ± 0.05 ^a^	0.34 ± 0.04 ^ab^	0.30 ± 0.08 ^ab^	0.27 ± 0.03 ^ab^	0.26 ± 0.05 ^b^	0.34 ± 0.06 ^ab^	0.23 ± 0.02 ^b^	0.26 ± 0.04 ^ab^	1508	1502
Furans											
2-Ethylfuran	0.89 ± 0.09	0.83 ± 0.07	0.90 ± 0.02	0.85 ± 0.05	0.84 ± 0.06	0.83 ± 0.02	0.89 ± 0.05	0.81 ± 0.03	0.92 ± 0.06	944	945
2-n-Butylfuran	0.96 ± 0.11	0.85 ± 0.08	0.82 ± 0.06	0.78 ± 0.04	0.90 ± 0.05	0.97 ± 0.09	0.86 ± 0.06	0.85 ± 0.07	0.92 ± 0.08	1134	1126
2-Pentylfuran	26.78 ± 1.25	25.19 ± 1.65	27.40 ± 2.48	27.17 ± 1.05	27.10 ± 1.59	25.80 ± 0.87	26.40 ± 2.09	25.57 ± 0.84	26.74 ± 2.14	1220	1226
2,3-Dihydrobenzofuran	1.00 ± 0.04 ^abc^	0.94 ± 0.03 ^abc^	0.86 ± 0.10 ^bc^	1.02 ± 0.08 ^ab^	0.98 ± 0.05 ^abc^	0.95 ± 0.07 ^abc^	1.05 ± 0.06 ^a^	0.99 ± 0.05 ^abc^	0.85 ± 0.04 ^c^	2292	NF
Alcohols											
2,3-Dimethylcyclohexanol	0.24 ± 0.05	0.22 ± 0.03	0.24 ± 0.04	0.21 ± 0.02	0.27 ± 0.01	0.23 ± 0.04	0.26 ± 0.03	0.25 ± 0.03	0.20 ± 0.02	1058	NF
1-Hexanol	0.87 ± 0.07 ^bc^	0.93 ± 0.09 ^ab^	0.78 ± 0.05 ^bc^	0.83 ± 0.04 ^bc^	0.81 ± 0.06 ^bc^	0.86 ± 0.03 ^bc^	0.73 ± 0.05 ^c^	1.04 ± 0.06 ^a^	0.85 ± 0.02 ^bc^	1355	1354
Ketones											
2-Heptanone	0.33 ± 0.05 ^bcd^	0.30 ± 0.03 ^d^	0.45 ± 0.05 ^ab^	0.34 ± 0.03 ^bcd^	0.44 ± 0.05 ^abc^	0.38 ± 0.02 ^abcd^	0.32 ± 0.04 ^cd^	0.39 ± 0.08 ^abcd^	0.49 ± 0.05 ^a^	1185	1183
2,2,6-Trimethylcyclohexanone	0.20 ± 0.04 ^ab^	0.15 ± 0.03 ^b^	0.23 ± 0.04 ^a^	0.15 ± 0.02 ^ab^	0.22 ± 0.04 ^ab^	0.17 ± 0.03 ^ab^	0.18 ± 0.01 ^ab^	0.17 ± 0.02 ^ab^	0.18 ± 0.03 ^ab^	1288	1282
2,3-Octanedione	1.46 ± 0.16	1.29 ± 0.12	1.34 ± 0.05	1.40 ± 0.08	1.26 ± 0.07	1.29 ± 0.11	1.29 ± 0.03	1.50 ± 0.07	1.44 ± 0.03 ^a^	1316	1325
6-Methyl-5-hepten-2-one	0.48 ± 0.07 ^abc^	0.31 ± 0.03 ^d^	0.37 ± 0.04 ^bcd^	0.39 ± 0.11 ^bcd^	0.52 ± 0.05 ^ab^	0.36 ± 0.03 ^cd^	0.55 ± 0.04 ^a^	0.49 ± 0.04 ^abc^	0.33 ± 0.03 ^d^	1330	1341
5-Ethyl-6-methyl-3E-hepten-2-one	1.96 ± 0.10	1.86 ± 0.09	2.10 ± 0.14	1.85 ± 0.05	2.02 ± 0.17	1.99 ± 0.16	1.90 ± 0.07	2.01 ± 0.10	1.88 ± 0.06	1443	NF
6,10-Dimethyl-5,9-Undecadien-2-one	0.80 ± 0.07	0.91 ± 0.11	0.77 ± 0.03	0.89 ± 0.06	0.79 ± 0.06	0.86 ± 0.02	0.93 ± 0.04	0.89 ± 0.09	0.90 ± 0.07	1838	1865
Others											
2-Acetylthiazole	0.12 ± 0.02 ^ab^	0.09 ± 0.01 ^b^	0.13 ± 0.02 ^ab^	0.14 ± 0.01 ^ab^	0.12 ± 0.02 ^ab^	0.15 ± 0.03 ^a^	0.11 ± 0.02 ^ab^	0.13 ± 0.02 ^ab^	0.16 ± 0.02 ^a^	1621	1643
Cyclopentyl 4-ethylbenzoate	0.31 ± 0.02 ^ab^	0.42 ± 0.03 ^a^	0.28 ± 0.06 ^b^	0.29 ± 0.07 ^ab^	0.22 ± 0.03 ^b^	0.34 ± 0.04 ^ab^	0.31 ± 0.07 ^ab^	0.24 ± 0.05 ^b^	0.25 ± 0.02 ^ab^	1798	1835
Phenol	0.06 ± 0.02 ^b^	0.07 ± 0.01 ^ab^	0.10 ± 0.01 ^a^	0.05 ± 0.01 ^b^	0.08 ± 0.02 ^ab^	0.07 ± 0.01 ^ab^	0.08 ± 0.01 ^ab^	0.06 ± 0.02 ^b^	0.09 ± 0.01 ^ab^	1964	2002
2-Methoxy-4-vinylphenol	0.89 ± 0.08	0.86 ± 0.03	0.91 ± 0.06	1.06 ± 0.10	0.93 ± 0.05	0.94 ± 0.11	1.02 ± 0.08	0.90 ± 0.12	0.87 ± 0.06	2146	2212
Indole	0.41 ± 0.05	0.43 ± 0.08	0.38 ± 0.02	0.45 ± 0.04	0.42 ± 0.07	0.40 ± 0.04	0.49 ± 0.07	0.51 ± 0.04	0.52 ± 0.14	2438	2450

Means followed by different lowercase letters within rows are significantly different (*p* < 0.05). Rows without letters indicate that there is no significant difference. CRI, calculated retention index; LRI, literature retention index, ND, not determined; NF, no literature found.

**Table 2 foods-11-03699-t002:** Phenolics profile of uncooked and cooked brown rice under different soaking conditions during cooking.

Phenolic Compound	Form	UC	Cooked Brown Rice under Different Soaking Conditions during Cooking								
			US	40-30	40-60	50-30	50-60	60-30	60-60	70-30	70-60
Ferulic acid (ng/g DW)	Bound	55.26 ± 1.22 ^a^	40.08 ± 0.75 ^b^	40.54 ± 0.88 ^b^	40.96 ± 0.27 ^b^	40.88 ± 0.66 ^b^	41.54 ± 1.08 ^b^	40.87 ± 0.54 ^b^	41.98 ± 0.74 ^b^	40.58 ± 0.97 ^b^	40.57 ± 0.75 ^b^
	Total	58.21 ± 0.97 ^a^	41.62 ± 0.94 ^b^	42.13 ± 0.58 ^b^	42.51 ± 0.35 ^b^	42.36 ± 0.74 ^b^	43.07 ± 1.04 ^b^	42.43 ± 0.47 ^b^	43.57 ± 0.64 ^b^	42.21 ± 0.78 ^b^	42.13 ± 0.89 ^b^
Isoferulic acid (ng/g DW)	Bound	14.28 ± 0.33 ^a^	12.56 ± 0.24 ^b^	12.94 ± 0.12 ^b^	11.78 ± 0.15 ^b^	12.65 ± 0.28 ^b^	12.84 ± 0.39 ^b^	12.51 ± 0.18 ^b^	13.14 ± 0.08 ^b^	12.95 ± 0.20 ^b^	12.43 ± 0.42 ^b^
	Total	14.28 ± 0.33 ^a^	12.56 ± 0.24 ^b^	12.94 ± 0.12 ^b^	11.78 ± 0.15 ^b^	12.65 ± 0.28 ^b^	12.84 ± 0.39 ^b^	12.51 ± 0.18 ^b^	13.14 ± 0.08 ^b^	12.95 ± 0.20 ^b^	12.43 ± 0.42 ^b^
*p*-Coumaric acid (ng/g DW)	Bound	10.60 ± 0.40 ^a^	9.10 ± 0.12 ^b^	8.92 ± 0.05 ^b^	9.17 ± 0.10 ^b^	9.12 ± 0.05 ^b^	9.05 ± 0.04 ^b^	9.00 ± 0.11 ^b^	9.13 ± 0.13 ^b^	9.14 ± 0.04 ^b^	9.04 ± 0.10 ^b^
	Total	14.08 ± 0.65 ^a^	10.22 ± 0.24 ^b^	10.11 ± 0.10 ^b^	10.39 ± 0.09 ^b^	10.29 ± 0.14 ^b^	10.20 ± 0.15 ^b^	10.24 ± 0.07 ^b^	10.20 ± 0.19 ^b^	10.28 ± 0.04 ^b^	10.21 ± 0.16 ^b^
Salicylic acid (ng/g DW)	Bound	1.59 ± 0.07 ^a^	1.39 ± 0.06 ^b^	1.42 ± 0.03 ^b^	1.40 ± 0.01 ^b^	1.39 ± 0.04 ^b^	1.37 ± 0.05 ^b^	1.38 ± 0.02 ^b^	1.39 ± 0.02 ^b^	1.38 ± 0.04 ^b^	1.40 ± 0.03 ^b^
	Total	1.59 ± 0.07 ^a^	1.39 ± 0.06 ^b^	1.42 ± 0.03 ^b^	1.40 ± 0.01 ^b^	1.39 ± 0.04 ^b^	1.37 ± 0.05 ^b^	1.38 ± 0.02 ^b^	1.39 ± 0.02 ^b^	1.38 ± 0.04 ^b^	1.40 ± 0.03 ^b^
Sum (ng/g DW)	Bound	81.73 ± 1.56 ^a^	63.13 ± 0.94 ^b^	63.82 ± 1.14 ^b^	63.31 ± 0.87 ^b^	64.04 ± 1.05 ^b^	64.80 ± 1.54 ^b^	63.76 ± 1.36 ^b^	65.64 ± 0.85 ^b^	66.05 ± 1.02 ^b^	63.44 ± 0.98 ^b^
	Total	88.16 ± 1.78 ^a^	65.79 ± 1.07 ^b^	66.60 ± 1.28 ^b^	66.08 ± 0.97 ^b^	66.69 ± 1.34 ^b^	67.48 ± 1.97 ^b^	66.56 ± 1.49 ^b^	68.30 ± 1.89 ^b^	68.82 ± 1.12 ^b^	66.16 ± 1.16 ^b^

Means followed by different lowercase letters within rows are significantly different (*p* < 0.05). ND, not determined. Abbreviation was listed: UC (uncooked brown rice), US (unsoaked brown rice); soaking conditions such as “40-30” respected soaking at 40 °C for 30 min.

## Data Availability

The datasets generated for this study are available on request to the corresponding author.

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
