# Peer review of "Effects of Soaking on the Volatile Compounds, Textural Property, Phytochemical Contents, and Antioxidant Capacity of Brown Rice"

_foods, 2022, doi:10.3390/foods11223699_

Round 1

Reviewer 1 Report

The draft is in line with the current literature supporting the interest of investigating some alternative cooking process and device to further optimize the quality of rice, and to favor energy-efficient cooking.

It is true that processing and consuming brown rice may help to get some healthier food but requires a longer time for cooking. The present manuscript aims at investigating the effect of soaking and/or cooking onto the instrumental texture of rice, onto the volatile compounds and some phytochemical and antioxidant polyphenol indicators after brown-rice cooking using a temperature/time adjustable rice cooking device. The introduction is documented to justify the interest of the investigation. An undeniable experimental work has been done to try to bring an original light on the topic. The authors claim to provide “a solid theoretical foundation to optimize automated brown rice cooking protocols”.

However, from the early beginning of the introduction, many approximations, short-cuts and missing information from the literature did not help to justify the experimental choice and methods used, to reinforce the interest and the originality of the current investigation.

In the result and discussion section, the figures and tables are often not suitable to support the overall poor discussion. Given the format (and size) of some data, some results cannot be highlighted. The authors should explicitly report what are the original results/innovation provided by the current investigation. Thus, it will help to later practically (and objectively) conclude in term of perspectives for the current investigation. What are the final recommendations and perspectives?

In the following section, some specific comments are made raw by raw. The authors are request to address all comments and questions in order to try to improve the overall quality of the current draft.

Specific comments to the authors

Title

The authors should revise the title. The authors need to mention that the investigation is designed using a rice cooker with an adjustable temperature profiling. The sensory quality was not evaluated, but the volatile compounds were.

 Abstract

Raws 46 Prior to the current paragraph, the authors should report here what are the current unit operations and devices available in the literature for the cooking process of rice, and report what difference it makes onto the palatability of paddy and brown rice (including textural, physicochemical and nutritional relevant traits). It is also relevant here to report what the key stages that contribute to the acceptability of cooked rice.

Raws 56 Figure 1 was not provided

Raws 57-59 Household rice cooker are not designed to provide any optimal soaking and cooking conditions, in order to cope with consumer demand and expectations. The consumers need to empirically define their optimal processing conditions if the device is designed for it.

Raws 61 “texture and volatile compounds are two critical..”

Raws 61 The authors should justify how the texture is usually experimentally evaluated in the literature and what are the current limitations of such approach in relation to the perception some trained panelists may have.

Raws 66 Need to be explicitly justified

Raws 70-71 I do not understand what the authors mean. What are the needed information ? How more sensitive detection methods can help to the cooking profiles, and to get a better quality of cooked brown rice?

Raws 76-77 Just speculation

 Materials and methods section

Raws 84 ABTS to be defined

Raws 94-97 to be rewritten

Raws 98-101 The authors should mention in the literature review the usual heating rates applied according to the cooking process. How do the authors justify the defined heating rate of un/soaked kernels ? It is already well-known from the literature that the initial heating stage is essential.

Raws 99-101 The authors should also justify the temperature monitoring using thermocouples and also provide some relevant data in the result and discussion section.

Raws 107-112 The experimental conditions used have to be mentioned (sample holding position, pre and post-test speed, relaxation time between cycles. Both hardness and stickiness parameters to be defined with relevant units. Stickiness computation is later wrong in the result and discussion section according to the reported unit.

Raws 111 What does “50% compression” mean ?

Raws 114 To be rewritten

Raws 115 the method has to be reported here

Raws 118-119 same comment as above.

Raws 123 the alkali hydrolysis method has to be reported here

Raws 153 the dilution of sodium persulfate is not mentioned

Raws 156 et 157-158 the methods have to be reported here

Results and discussion section

All labels and numbers have to be checked and corrected in the tables, figures and citations.

Raws 166-167 Is there any synergistic effect?

Raws 168-169 off topic

Figure 1 the X-axis should be added and a legend to recall on A-B values (for soaking treatments) and “US” label. The Y-axis unit has to be corrected according to the usual rheological units for stickiness parameter. The authors are advised to include some letters to illustrate the significant differences among treatments.

Raws 173-174 units to be corrected

Raws 174-175 to be rewritten

Raws 177 “relates” but not “correlates strongly”

Raws 184-186 How much time is needed to observe the phenomenon?

Raws 188 Did the stickiness significantly decreases?

Raws 188-190 to be rewritten

Figure 2 The different chemical groups have to be clearly defined in the material and method section

Raws 199-200 to be rewritten

Raws 207-210 the argument should be revised and tempered. The texture is off topic here. The fact that no additional undesirable volatile compounds were observed, does not mean that a sensory evaluation might be optimal since some interaction between estimated volatile compounds may give an unfavorable flavor, that could only be estimated by sensory evaluation.

Table 1 is not readable. The authors should only report the significant differences among treatments and compounds. One solution could be to provide the full data as supplementary data and only report significant differences with “*” and “**), according to the level of significance.

Figure 3A, significant differences among treatments to be illustrated. X axis label to be added and a figure caption made.

Figure 3B and C to be combined using double Y axis. X axis label to be added (+ figure caption).

Table 2 “Free ferulic acid and coumaric acid raws to be removed. Again, Table 2 is not easily readable. The authors should only report the significant differences among treatments and compounds. One solution could be to provide the full data as supplementary data and only report significant differences with “*” and “**), according to the level of significance.

Figure 4 . Both A and B figures should be combined. Both free and bound ABTS and FRAP should also appear under a unique histogram illustrating the different total amounts (each rectangle containing both sub-rectangles of free and bound ABTS amounts, and separately both free and bound FRAP amounts).

Raws 297-312 The authors should later compare the significant differences observe between both ABTS and FRAP determination and discuss it.

Conclusions

Raws 314-316 to be rewritten with no bias.

Raws 316-321 An objective summary should be done while reporting the most significant results obtained in the different sub-sections.

Raws 322-325 The sentence must be revised. What are the practical advice to be delivered according to results above? Does it help to drive the cooking process using a set temperature/time in a rice cooker to get an optimal textural, phytochemical or antioxidant capacity in brown rice?

Author Response

Dear Editors and Reviewers,

We really appreciate your helpful comments and valuable suggestions. Those comments are all valuable for improving our paper and also make us analyse information more scientifically. We have thoroughly considered all the comments and substantially revised our manuscript. The point-to-point answers and explanations for all revisions were listed in the following.

We hope, with these modifications and improvements based on the comments of the reviewers, the quality of our manuscript would meet the publication standard.

Yours sincerely,

Hui Zhang

Jiangnan University

State Key Laboratory of Food Science and Technology & School of Food Science and Technology, Jiangnan University, 1800 Lihu Avenue, Wuxi 214122, China

Review 1:
The draft is in line with the current literature supporting the interest of investigating some alternative cooking process and device to further optimize the quality of rice, and to favor energy-efficient cooking.

It is true that processing and consuming brown rice may help to get some healthier food but requires a longer time for cooking. The present manuscript aims at investigating the effect of soaking and/or cooking onto the instrumental texture of rice, onto the volatile compounds and some phytochemical and antioxidant polyphenol indicators after brown-rice cooking using a temperature/time adjustable rice cooking device. The introduction is documented to justify the interest of the investigation. An undeniable experimental work has been done to try to bring an original light on the topic. The authors claim to provide “a solid theoretical foundation to optimize automated brown rice cooking protocols”.

However, from the early beginning of the introduction, many approximations, short-cuts and missing information from the literature did not help to justify the experimental choice and methods used, to reinforce the interest and the originality of the current investigation.

In the result and discussion section, the figures and tables are often not suitable to support the overall poor discussion. Given the format (and size) of some data, some results cannot be highlighted. The authors should explicitly report what are the original results/innovation provided by the current investigation. Thus, it will help to later practically (and objectively) conclude in term of perspectives for the current investigation. What are the final recommendations and perspectives?

In the following section, some specific comments are made raw by raw. The authors are request to address all comments and questions in order to try to improve the overall quality of the current draft.

Specific comments to the authors

Title

The authors should revise the title. The authors need to mention that the investigation is designed using a rice cooker with an adjustable temperature profiling. The sensory quality was not evaluated, but the volatile compounds were.

Response: Thank you for your comment. The title has been revised.

“Effects of Soaking on the Volatile Compounds, Textural Property, Phytochemical Contents, and Antioxidant Capacity of Brown Rice” (Line 2).

Abstract

Raws 46 Prior to the current paragraph, the authors should report here what are the current unit operations and devices available in the literature for the cooking process of rice, and report what difference it makes onto the palatability of paddy and brown rice (including textural, physicochemical and nutritional relevant traits). It is also relevant here to report what the key stages that contribute to the acceptability of cooked rice.

Response: Your comment is very helpful. Additional information has been added in revised manuscript. (Line 47-53)

Raws 56 Figure 1 was not provided

Response: Sorry for the mistake. Initially, Fig. 1 was used to show the cooking curves of different brands of rice cookers (mixed with each other). It has been deleted in revised manuscript.

Raws 57-59 Household rice cooker are not designed to provide any optimal soaking and cooking conditions, in order to cope with consumer demand and expectations. The consumers need to empirically define their optimal processing conditions if the device is designed for it.

Response: Sorry for the unclear description. It is true that household rice cookers are not designed to provide any optimal soaking and cooking conditions. Different consumers have different preferences. In this part, we intend to express that different rice cookers have different soaking conditions, but the selecting rules or relationships between properties of cooked rice and soaking conditions are still unclear.

Raws 61 “texture and volatile compounds are two critical..”

Response: In the revised manuscript, “texture and volatile compounds are the two critical.” has been changed to “texture and volatile compounds are two critical.”

Raws 61 The authors should justify how the texture is usually experimentally evaluated in the literature and what are the current limitations of such approach in relation to the perception some trained panelists may have.

Response:We have taken your suggestion into consideration. Additional information has been added in revised manuscript as follows: “Instead of sensory evaluation by trained panelists, instrumental measurement of texture is often used to evaluate the mechanical characteristics of foods during the oral processing. The texture profile analysis (TPA) has been used to develop to understand the texture properties of different foods and the texture terms are classified into five basic parameters. For cooked rice, hardness and stickiness are two key indexes for the palatability evaluation.” (Line 69-74)

Raws 66 Need to be explicitly justified

Response: Sorry, we did not express this part clearly. In revised manuscript Response: “Unfortunately, these soaking conditions (long time soaking and refilling water) are not suitable when using a rice cooker for convenience and effective cooking”. (Line 78-80)

Raws 70-71 I do not understand what the authors mean. What are the needed information ? How more sensitive detection methods can help to the cooking profiles, and to get a better quality of cooked brown rice?

Response: Your suggestion is very helpful, more clear expression has been added in revised manuscript. “More information and more sensitive detection methods are also needed to optimize the cooking curves to obtain better quality of cooked brown rice.” has been changed to “More sensitive cooking device and more quality comparison information under uniform hardware conditions are also needed to optimize the cooking curves to obtain better quality of cooked brown rice.” (Line 83-86)

Raws 76-77 Just speculation

Response: “Our study provides a solid theoretical foundation to optimize automated brown rice-cooking protocols.” has been changed to “Our study tries to provide a  theoretical foundation to optimize automated brown rice-cooking protocols.” (Line 91-92)

Materials and methods section

Raws 84 ABTS to be defined

Response: ABTS has been defined in revised manuscript. (Line 100)

Raws 94-97 to be rewritten

Response: Thanks for your comment. The description has been rewritten as follows “Briefly, 400g of brown rice and 800 mL distilled water were placed into the adjustable rice cooker. Then, eight cooking processes with different soaking temperatures (40, 50, 60 and 70 °C) and time (30 and 60 min) were used. After soaking, the temperature increased to 100 °C and maintaining it for 40 min.” (Line 109-112)

Raws 98-101 The authors should mention in the literature review the usual heating rates applied according to the cooking process. How do the authors justify the defined heating rate of un/soaked kernels ? It is already well-known from the literature that the initial heating stage is essential.

Response: Thank you for your comments. The setting of soaking conditions and heating rate was based on the data collected from different brands of rice cooker (several of them was shown below). The literature reported about heating rate or cooking curves was rare, and some mainly depended on the heating device (aluminum pan) in the laboratory or rice cooker with unchangeable program.

The disclosure of all cooking data of different brands of rice cookers was restricted, so we cannot provide it.

Raws 99-101 The authors should also justify the temperature monitoring using thermocouples and also provide some relevant data in the result and discussion section.

Response: The aim of real-time temperature monitoring was to collect the temperature change process of different brands of rice cookers and also check the stability of temperature of adjustable rice during rice cooking. As mentioned above, the cooking data of these rice cookers were restricted, thus we cannot provide them in the result and discussion. (Line 115-118)

Raws 107-112 The experimental conditions used have to be mentioned (sample holding position, pre and post-test speed, relaxation time between cycles. Both hardness and stickiness parameters to be defined with relevant units. Stickiness computation is later wrong in the result and discussion section according to the reported unit.

Response: Thank you for your comment, a detail description of the texture method has been provided. (Line 124-129)

Raws 111 What does “50% compression” mean ?

Response: “50% compression” means “the deformation of cooked rice was kept at 50% of the total compression”. (Line 129)

Raws 114 To be rewritten

Response: The method has been rewritten. (Line 130-145)

Raws 115 the method has to be reported here

Response: Detail information about the method has been provided in revised manuscript. (Line 130-145)

Raws 118-119 same comment as above.

Response: Your comments are very helpful. A detail description has been provided in the revised manuscript. (Line 146-149)

Raws 123 the alkali hydrolysis method has to be reported here

Response: Your comments are very helpful. Additional information has been added in the revised manuscript. (Line 153-154)

Raws 153 the dilution of sodium persulfate is not mentioned

Response: The preparation of ABTS solution has been rewritten. (Line 182-183)

Raws 156 et 157-158 the methods have to be reported here

Response: The methods have been reported in revised manuscript. (Line 188-192)

Results and discussion section

All labels and numbers have to be checked and corrected in the tables, figures and citations.

Response: Thank you for your comment, all labels and numbers have been checked.

Raws 166-167 Is there any synergistic effect?

Response: The expression maybe was not accurate enough. In revised manuscript “A higher soaking temperature or a longer period of soaking could promote greater water absorption by the endosperm throughout the outer bran layer.” has been changed (Line 198-202)

Raws 168-169 off topic

Response: Thanks for your comment. We intended to express that the lower hardness was related with the water content and fully gelatinization of starch granules. The expression has been revised more clearly: “The increased moisture content and its even distribution could facilitate the granules swelling and gelatinization of starch upon heating, and thus exhibited a lower hardness of cooked rice [23, 24].” (Line 198-202)

Figure 1 the X-axis should be added and a legend to recall on A-B values (for soaking treatments) and “US” label. The Y-axis unit has to be corrected according to the usual rheological units for stickiness parameter. The authors are advised to include some letters to illustrate the significant differences among treatments.

Response: Your suggestions are very helpful. We have taken your suggestions into consideration. The figure caption has been revised. The usual units of hardness (maximum positive peak force) and stickiness (negative peak area) were g, N and g.sec, N.sec, respectively. In this paper, we choose g and g.sec as the units of hardness and stickiness on the recommendation of the engineer and reference (Li, H.; Prakash, S.; Nicholson, T. M.; Fitzgerald, M. A.; Gilbert, R. G., The importance of amylose and amylopectin fine structure for textural properties of cooked rice grains. Food Chem. 196, 702-711). (Line 204)

The values and significant differences among treatments were provided in supplementary material (Table S1) as combined with the comments of another reviewer.

Table S1 The texture properties of cooked brown rice under different soaking conditions

Samples

Hardness (g)

Stickiness (g.sec

Unsoaked rice

2033±84a

176.5±29.7f

Soaking at different conditions

40°C-30min

2001±96ab

 374.1±39.5ab

40°C-60min

1975±95ab

397.2±25.4a

50°C-30min

1995±93ab

305.0±27.6cd

50°C-60min

1846±73cd

 387.0±36.4a

60°C-30min

1916±75bc

287.4±28.7de

60°C-60min

1734±85de

344.6±27.2bc

70°C-30min

1816±78cd

255.6±21.1e

70°C-60min

1639±50e

293.8±27.7d

Means followed by different lowercase letters within rows are significantly different (p <0.05).

Raws 173-174 units to be corrected

Response: The units have been checked and revised. (Line 209-210)

Raws 174-175 to be rewritten

Response: The expression has been rewritten. (Line 210-212)

Raws 177 “relates” but not “correlates strongly”

Response: “correlates strongly” has been changed to  “relates”. (Line 217)

Raws 184-186 How much time is needed to observe the phenomenon?

Response: As mentioned in the literature this phenomenon was observe within 30 min.

Raws 188 Did the stickiness significantly decreases?

Response: Thank you for your comment. The conclusion has been revised.

Raws 188-190 to be rewritten

Response: Thank you for your comment. The conclusion has been rewritten. (Line 226-232)

Figure 2 The different chemical groups have to be clearly defined in the material and method section

Response: Your suggestion is very helpful. The detail information about different chemical groups has been mentioned in the method section in reversed manuscript.  (Line 130-145).

Raws 199-200 to be rewritten

Response: “In general, germinated brown rice treated with soaking at room temperature for hours might develop an undesirable flavor.” has been changed to “As mentioned in literature, germinated brown rice might develop undesirable flavors when soaked at room temperature for hours”. (Line 235-236)

Raws 207-210 the argument should be revised and tempered. The texture is off topic here. The fact that no additional undesirable volatile compounds were observed, does not mean that a sensory evaluation might be optimal since some interaction between estimated volatile compounds may give an unfavorable flavor, that could only be estimated by sensory evaluation.

Response: Your suggestion is very pertinent. The conclusion has been changed to “The contents of these volatile compounds were similar and no new flavor substances were produced, indicating that a certain degree of soaking did not introduce undesirable flavors.”. (Line 245-246)

Table 1 is not readable. The authors should only report the significant differences among treatments and compounds. One solution could be to provide the full data as supplementary data and only report significant differences with “*” and “**), according to the level of significance.

Response:  Thanks for your suggestion. It is clear to report the significant differences with “*” and “**” if the groups are no more than three. In our research, yet, there are nine groups, including eight treated samples and the control, were compared between any two groups. It would be unclear if only “*” and “**” are marked. It will be more clear and more intuitional to use the letters to represent the significance after thinking twice.

Figure 3A, significant differences among treatments to be illustrated. X axis label to be added and a figure caption made.

Response: Thanks for your suggestion. Figure 3 has been revised. (line 325-330)

Figure 3B and C to be combined using double Y axis. X axis label to be added (+ figure caption).

Response: Thanks for your suggestion. Figure 3has been revised. (Line 325-330)

Table 2 “Free ferulic acid and coumaric acid raws to be removed. Again, Table 2 is not easily readable. The authors should only report the significant differences among treatments and compounds. One solution could be to provide the full data as supplementary data and only report significant differences with “*” and “**), according to the level of significance.

Response: It is the same with Table 1. Table 2 is a little more complicated than Table 1, so the table we represented in the original manuscript is better.

Figure 4 . Both A and B figures should be combined. Both free and bound ABTS and FRAP should also appear under a unique histogram illustrating the different total amounts (each rectangle containing both sub-rectangles of free and bound ABTS amounts, and separately both free and bound FRAP amounts).

Response: Your suggestion is very helpful. A new figure 4 has been presented in revised manuscript. (Line 345)

Raws 297-312 The authors should later compare the significant differences observe between both ABTS and FRAP determination and discuss it.

Response: Your suggestion is very helpful. A comparison of the significant difference has been added in revised manuscript. (Line 351-355)

Conclusions

Raws 314-316 to be rewritten with no bias.

Response: This section has been rewritten. (Line 360-373)

Raws 316-321 An objective summary should be done while reporting the most significant results obtained in the different sub-sections.

Response: Thanks for your comment, and this section has been rewritten. (Line 360-373)

Raws 322-325 The sentence must be revised. What are the practical advice to be delivered according to results above? Does it help to drive the cooking process using a set temperature/time in a rice cooker to get an optimal textural, phytochemical or antioxidant capacity in brown rice?

Response: Your comment is very useful, we have taken it into consideration, the conclusion has been revised manuscript. (Line 360-373)

Reviewer 2 Report

Units are not given in figure 1

Figure 1A, where hardness values are given, is not given.

Line 175, Fig 2 should be Fig 1

Figure 2 has to be relocated  

It is not clear what the columns in Figure 2 are, they need to be changed.

The location and units of all figures should be checked and changed.       

There are many mistakes in the table 2.Table 2 should be examined carefully and necessary corrections should be made.

How can the total amount of isoferulic acid be 14.28±0.33 while the ND in the Free form is 14.28±0.33 in bound?

Although antioxidant properties of phenolic compounds are given in Figure 4, why is it expressed as Figure 5?        

Author Response

Dear Editors and Reviewers,

We really appreciate your helpful comments and valuable suggestions. Those comments are all valuable for improving our paper and also make us analyse information more scientifically. We have thoroughly considered all the comments and substantially revised our manuscript. The point-to-point answers and explanations for all revisions were listed in the following.

We hope, with these modifications and improvements based on the comments of the reviewers, the quality of our manuscript would meet the publication standard.

Yours sincerely,

Hui Zhang

Jiangnan University

State Key Laboratory of Food Science and Technology & School of Food Science and Technology, Jiangnan University, 1800 Lihu Avenue, Wuxi 214122, China

Review 2:

Units are not given in figure 1

Response: Units has been added.

Figure 1A, where hardness values are given, is not given.

Response: Figure 1A has been presented in revised manuscript. (Line 204)

Line 175, Fig 2 should be Fig 1

Response: The mistake has been corrected. (Line 212)

Figure 2 has to be relocated  

Response: Figures were typeset by editors, and we have relocated the figures in revised manuscript according to your suggestions.

It is not clear what the columns in Figure 2 are, they need to be changed.

Response: Your suggestion is very helpful. Figure 2 has been changed. (Line 247)

The location and units of all figures should be checked and changed. 

Response: Your advice is very helpful. The location and units of figures have been checked and corrected.

There are many mistakes in the table 2.Table 2 should be examined carefully and necessary corrections should be made.

Response: Thanks a lot for your advice. Table 2 has been corrected in revised manuscript.

How can the total amount of isoferulic acid be 14.28±0.33 while the ND in the Free form is 14.28±0.33 in bound?

Response:The total amount of isoferulic acid was composed by the free form and bound form. Thus the total amount was the sum of free form (ND) and bound form (14.28±0.33). (Line 301-304)

Although antioxidant properties of phenolic compounds are given in Figure 4, why is it expressed as Figure 5?   

Response: The mistake has been corrected. “Figure 5” has been changed to “Figure 4”.  (Line 334, 344)

Reviewer 3 Report

The research proposes to answer knowledge gaps related to brown rice soaking. However, there are certain sections that can improve the impact and understanding of the work. I would like to point out the following:

Lines 40 to 45 need a reference.

Lines 72 to 77 are not reflected in the abstract and this minimizes the impact of the work.

Line 86 needs correction.

Lines 98 and 99 because the authors used different heating rates?

Lines 107 to 109 should be rewritten because it is repetitive.

Line 112 post test speed was the same?

Line 118 the flour refers to the freeze-dried cooked samples?

Lines 164 to 166 it talks about hardness, but the graph shows stickiness? please revise.

Line 170 the graph is incomplete, only section B appears. Also, the title of the figure should be on the same sheet.

Lines 173 to 190 the discussion requires more depth, the authors should discuss with respect to the fine structure of starch especially amylopectin.

Line 191 figure 2 and its title should be on the same page.

Line 211 the authors should improve the presentation of Table 1, the data are so tightly packed that it is difficult to read. Subsequently, the enormous amount of data obtained is grouped together (lines 215 to 231), the authors should consider the relevance of placing Table 1 or leave it as supplementary material.

The authors should present the information in order, first the discussion section and then the graphs and tables to support the discussion, not first the graphs and tables and then the discussion.

Line 322, what are the modifications of the soaking conditions?

Author Response

Dear Editors and Reviewers,

We really appreciate your helpful comments and valuable suggestions. Those comments are all valuable for improving our paper and also make us analyse information more scientifically. We have thoroughly considered all the comments and substantially revised our manuscript. The point-to-point answers and explanations for all revisions were listed in the following.

We hope, with these modifications and improvements based on the comments of the reviewers, the quality of our manuscript would meet the publication standard.

Yours sincerely,

Hui Zhang

Jiangnan University

State Key Laboratory of Food Science and Technology & School of Food Science and Technology, Jiangnan University, 1800 Lihu Avenue, Wuxi 214122, China

Review 3:  

The research proposes to answer knowledge gaps related to brown rice soaking. However, there are certain sections that can improve the impact and understanding of the work. I would like to point out the following:

Lines 40 to 45 need a reference.

Response: A reference has been added. (Line 43)

Lines 72 to 77 are not reflected in the abstract and this minimizes the impact of the work.

Response: Your suggestion is very helpful. additional information has been added in the abstract. (Line 16-17)

Line 86 needs correction.

Response: The format has been corrected. (Line 100-102)

Lines 98 and 99 because the authors used different heating rates?

Response: Thank you for your comments. The setting of soaking conditions and heating rate was based on the data collected from different brands of rice cooker. The soaked samples usually have higher heating rate to rapidly reach soaking conditions, while a lower heating rate of un-soaked sample provides enough time for rice cooking.

Lines 107 to 109 should be rewritten because it is repetitive.

Response: Your comment is very helpful. The texture measurement has been rewritten. (Line 124-129)

Line 112 post test speed was the same?

Response: Additional information about the method has been added. (Line 124-129)

Line 118 the flour refers to the freeze-dried cooked samples?

Response: Yes, the flour refers to the freeze-dried cooked samples.

Lines 164 to 166 it talks about hardness, but the graph shows stickiness? please revise.

Response: Figure 1A has been added. (Line 204)

Line 170 the graph is incomplete, only section B appears. Also, the title of the figure should be on the same sheet.

Response: Figure 1A has been added, and the title has been changed. (Line 204)

Lines 173 to 190 the discussion requires more depth, the authors should discuss with respect to the fine structure of starch especially amylopectin.

Response: The discussion about the relationship between stickiness and amylopectin has been added in revised manuscript. (Line 218-221)

Line 191 figure 2 and its title should be on the same page.

Response: The location of Figure 2 has been changed. (Line 247)

Line 211 the authors should improve the presentation of Table 1, the data are so tightly packed that it is difficult to read. Subsequently, the enormous amount of data obtained is grouped together (lines 215 to 231), the authors should consider the relevance of placing Table 1 or leave it as supplementary material.

Response: Thank you for your comment. Table 1 has been changed in revised manuscript. (Line 268)

The authors should present the information in order, first the discussion section and then the graphs and tables to support the discussion, not first the graphs and tables and then the discussion.

Response: The location of graphs and tables has been changed in revised manuscript.

Line 322, what are the modifications of the soaking conditions?

Response: Sorry, the statements are inaccurate. In revised manuscript “Therefore, we propose modifications to the parameters of soaking conditions for individual cooking curves in household rice cookers to achieve better sensory and nutritional quality of cooked brown rice.” has been changed to “This study help us and the manufacturer to drive the cooking process using a set temperature/time in a rice cooker to get an optimal textural, phytochemical and antioxidant capacity in brown rice.” (Line 370-373).

Round 2

Reviewer 1 Report

The authors are request to address all my comments/suggestions below.

Specific comments to the authors

Raws 113-114. The authors should mention that the chosen heating rate for both soaked and unsaoked rice sample were chosen according to usual condition applied by rice cooker brands.

Raws 114-116 The sentence should be removed from the manuscript, since no data is provided related to the temperature monitoring.

Raws 127 “the deformation of cooked rice was kept at 50% of the total compression”. A rising stress was applied to the sample until getting a maximum strain of 50% (with the estimation of the initial height of the sample taken into account). One might think that the authors were carrying out a stress relaxation test… Since a double cycle compression test was applied, the strain cannot be kept constant. Once again, the sentence is not clear and need to be rewritten.

Raws 130-145 The authors should avoid the term “our previously”

Raws 151-152 Sentence to be checked and revised

Raws 208-210, According to the figure, the correlations are not obvious.

Raws 223 “..a greater water gradient”

Raws 223 The authors should avoid “between them”

Raws 223-225 The sentence has to be revised and English checked.

Raws 226-227 The significant decrease of the stickiness is not obvious.

Raws 265 Table 1 needs to be improved. No need to provide any letter when no significant difference is observed between process (ie octane, pinene,..)

Raws 297 Table 2 need to be further improved. Useless free isoferulic acid, free coumaric acid, and free salicylic acic raws have to be removed.

Raws 346-350 The added paragraph seems relevant. However, one aspect was not taken into account after my previous review: the authors should also compare and discuss about the significant differences observed between ABTS and FRAP methods. If no comparison is made in between, the justification of the choice of the 2 methods in the material and method section as well as the corresponding results presented in the result and discussion section are not obvious. What is then the added-value to provide both data for both methods? A comparison and discussion is needed.

Author Response

Dear Editors and Reviewers,

We really appreciate your helpful comments and valuable suggestions. We have thoroughly considered all the comments and substantially revised our manuscript. Revised portion were marked in RED in the manuscript. The point-to-point answers and explanations for all revisions were listed in the following.

We hope, with these modifications and improvements based on the comments of the reviewers, the quality of our manuscript would meet the publication standard.

Yours sincerely,

Hui Zhang

Jiangnan University

State Key Laboratory of Food Science and Technology & School of Food Science and Technology, Jiangnan University, 1800 Lihu Avenue, Wuxi 214122, China

Specific comments to the authors

Q1. Raws 113-114. The authors should mention that the chosen heating rate for both soaked and unsaoked rice sample were chosen according to usual condition applied by rice cooker brands.

Response: Your suggestion is helpful. Additional information about heating rate has been added in revised manuscript (Line 115-116).

Q2. Raws 114-116 The sentence should be removed from the manuscript, since no data is provided related to the temperature monitoring.

Response: The sentence has been deleted.

Q3. Raws 127 “the deformation of cooked rice was kept at 50% of the total compression”. A rising stress was applied to the sample until getting a maximum strain of 50% (with the estimation of the initial height of the sample taken into account). One might think that the authors were carrying out a stress relaxation test… Since a double cycle compression test was applied, the strain cannot be kept constant. Once again, the sentence is not clear and need to be rewritten.

Response:Your suggestion is very helpful, the description has been revised in manuscript (Line 127-128). 

Q4. Raws 130-145 The authors should avoid the term “our previously”

Response: We have taken your suggestion into consideration (Line 130).

Q5. Raws 151-152 Sentence to be checked and revised

Response: The sentence has been revised (Line 150-152).

Q6. Raws 208-210, According to the figure, the correlations are not obvious.

Response: Thank you for your comment. A clear description has been presented in revised manuscript (Line 208-210). By comparison, the stickiness of cooked rice soaked at 60 min was higher than that at 30 min (at the same soaking temperature), and it was decreased with the soaking temperature at the same soaking time. Thus, we concluded as follows: “Further, the comparison of different soaking temperatures and times showed that the stickiness correlated positively with the soaking time (the same soaking temperatures) and negatively with the soaking temperature (the same soaking times) (Fig. 1).”

Q7. Raws 223 “..a greater water gradient”

Response: The sentence has been revised (Line 224).

Q8. Raws 223 The authors should avoid “between them”

Response: We have taken your suggestion into consideration (Line 225).

Q9. Raws 223-225 The sentence has to be revised and English checked.

Response: The sentences have been rewritten (Line 225).

Q10. Raws 226-227 The significant decrease of the stickiness is not obvious.

Response: The description has been changed (Line 228).

Q11. Raws 265 Table 1 needs to be improved. No need to provide any letter when no significant difference is observed between process (ie octane, pinene,..)

Response: Your comment is very helpful, table 1 has been improved in revised manuscript (Line 267).

Q12. Raws 297 Table 2 need to be further improved. Useless free isoferulic acid, free coumaric acid, and free salicylic acic raws have to be removed.

Response: Your comment is very helpful, table 2 has been improved in revised manuscript (Line 300).

Q13. Raws 346-350 The added paragraph seems relevant. However, one aspect was not taken into account after my previous review: the authors should also compare and discuss about the significant differences observed between ABTS and FRAP methods. If no comparison is made in between, the justification of the choice of the 2 methods in the material and method section as well as the corresponding results presented in the result and discussion section are not obvious. What is then the added-value to provide both data for both methods? A comparison and discussion is needed.

Response: Thank you for your comment, additional information has been added in revised manuscript.

Reviewer 3 Report

I appreciate the modifications made to the document for a better understanding of the research.

Author Response

We really appreciate your helpful comments and valuable suggestions.